# Differential sensing with arrays of de novo designed peptide assemblies

William M. Dawson [1,11] ✉, Kathryn L. Shelley [1,2,11], Jordan M. Fletcher[1,3], D. Arne Scott[1,2,3], Lucia Lombardi [1,4,5], Guto G. Rhys [1,6,7], Tania J. LaGambina [3], Ulrike Obst[3], Antony J. Burton[1,8], Jessica A. Cross [1,2], George Davies[1], Freddie J. O. Martin[1], Francis J. Wiseman[1], R. Leo Brady [2], David Tew [9], Christopher W. Wood[1,2,10] ✉ & Derek N. Woolfson [1,2,4] ✉

Differential sensing attempts to mimic the mammalian senses of smell and taste to identify analytes and complex mixtures. In place of hundreds of complex, membrane-bound G-protein coupled receptors, differential sensors employ arrays of small molecules. Here we show that arrays of computationally designed de novo peptides provide alternative synthetic receptors for differential sensing. We use self-assembling α-helical barrels (αHBs) with central channels that can be altered predictably to vary their sizes, shapes and chemistries. The channels accommodate environment-sensitive dyes that fluoresce upon binding. Challenging arrays of dye-loaded barrels with analytes causes differential fluorophore displacement. The resulting fluorimetric fingerprints are used to train machine-learning models that relate the patterns to the analytes. We show that this system discriminates between a range of biomolecules, drink, and diagnostically relevant biological samples. As αHBs are robust and chemically diverse, the system has potential to sense many analytes in various settings.

Mammalian olfaction—the sense of smell—discriminates between many odorant molecules[1]. It achieves this using 300–2000 G-protein coupled receptors (GPCRs)[2,3]. Rather than making specific receptor-odorant interactions, each receptor responds to a variety of molecules[4]. The composite response is interpreted by the brain as a smell. Differential sensing attempts to mimic this[5,6]. GPCRs are membrane-spanning proteins, making them difficult to manipulate. Indeed, attempts to use them in sensing have met with limited success[7,8]. Therefore, differential sensors employ various organic molecules and other moieties that interact with analytes in non-specific ways.

For example, current differential sensors use synthetic reporters or receptors, including: chemo-responsive pigments, metal nanoparticles and quantum dots, carbon nanotubes, metal oxides, and supramolecular or peptide-based systems[5,6]. In each case, arrays of the synthetic molecules are challenged with analytes, and electrical or optical readouts are analyzed chemometrically. In this way, systems that differentiate terpenes[9], fatty acids[10], amino acids[11] and sugars[12], amongst other biomolecules, have been developed. A strength of differential sensing over traditional biosensors that target a single defined analyte or biomarker is the potential to process and discriminate between complex mixtures of analytes. Accordingly,

[1]School of Chemistry, University of Bristol, Cantock's Close, Bristol BS8 1TS, UK. [2]School of Biochemistry, University of Bristol, Medical Sciences Building, University Walk, Bristol BS8 1TD, UK. [3]Rosa Biotech, Science Creates St Philips, Albert Road, Bristol BS2 0XJ, UK. [4]BrisSynBio, University of Bristol, School of Chemistry, Bristol BS8 1TS, UK. [5]Department of Chemical Engineering, Imperial College London, London SW7 2AZ, UK. [6]Department of Biochemistry, University of Bayreuth, Universitätsstraße 30, 95447 Bayreuth, Germany. [7]School of Chemistry, Cardiff University, Main Building, Park Place, Cardiff CF10 3AT, UK. [8]AstraZeneca, 35 Gatehouse Drive, Waltham, MA 02451, USA. [9]GlaxoSmithKline (GSK), Gunnels Wood Rd, Stevenage SG21 2NY, UK. [10]School of Biological Sciences, University of Edinburgh, Roger Land Building, Edinburgh EH9 3JQ, UK. [11]These authors contributed equally: William M. Dawson, Kathryn L. Shelley. ✉e-mail: w.dawson@bristol.ac.uk; chris.wood@ed.ac.uk; D.N.Woolfson@bristol.ac.uk

differential sensing has been used successfully in food-and-drink[13], pollutant-monitoring[14], biomedical[15], and national-security applications[16].

Although some natural proteins—including fluorescent proteins[17,18] and serum albumins[9,10]—have been used as the receptor components, proteins have yet to be fully exploited in differential sensing. De novo designed peptides and proteins are exciting prospects here, as their structures and chemistries can be tailored for specific purposes[19–21]. Although engineered and de novo proteins are being applied to sense targeted analytes[22–26], there are no reported uses of de novo proteins in differential sensing. We speculated that recently developed α-helical barrels (αHBs) would be promising candidates for this[27,28].

αHBs are oligomers of 5 or more α-helical peptides that assemble into coiled-coil structures with central solvent-accessible channels. Typically, the component peptides are ≈30 amino acids long with ≈8 of these lining the lumens. Therefore, the chemical space available to αHBs is large. Robust rational and computational methods have been developed to design αHBs[27,29,30]. These allow oligomer-state specification and, thus, the size and shape of the internal cavities to be controlled. Furthermore, the channel-facing side chains can be altered, which has allowed αHBs to be functionalized to make tubular biomaterials[31], catalysts[32], small-molecule binders[28], and membrane-spanning ion channels[33]. In these ways, αHBs are analogous to other natural and synthetic receptors: they are highly mutable helical bundles with the ability to bind a variety of substrates. However, αHBs are water soluble, thermally stable, and can be made at scale. Moreover, there are established sequence-to-structure relationships, or design rules, that allow αHBs to be constructed and engineered with confidence.

Here we demonstrate the utility of αHBs as components of a differential sensing platform (Fig. 1). This has an array of 46 αHBs spanning chemical and structural space. The αHBs are loaded with an environment sensitive dye (in this case, 1,6-diphenyl-1,3,5-hexatriene, DPH) that binds within the channels and fluoresces. The size and shape of the channel for each αHB dictates how strongly the dye binds to each assembly, and the affinity of analyte molecules that could potentially displace the dye. Accordingly, challenging the array with

analytes leads to differential displacement of the dye across the array to give a fluorescent fingerprint. These signals are interrogated by machine learning (ML) to relate the fingerprints to the analytes. We use various ML models to classify fingerprints and use them predictively for 15 different analytes from 3 types of biomolecules, and for complex mixtures including serological samples of non-alcoholic fatty liver disease (NAFLD). NAFLD is currently under diagnosed, demonstrating the potential of our system in medical in vitro diagnostics. Finally, the features that contribute to successful ML models reveal how the αHB array is analogous to other differential sensing technologies, and how the platform can be tailored to specific applications.

## Results and discussion
### Rational design delivers an array of α-helical barrels

To access a broad-spectrum of small-molecule binding and hence analyte sensing, we sought to construct an array of de novo designed αHBs with predictably varying sizes, shapes and chemistries of the internal channels. We reasoned that this should be possible because αHBs are hyperthermostable, tolerate mutations, and have well-established design rules[27,29]. We targeted two properties of αHBs: oligomeric state, which directly affects the internal diameter of the channel; and the identities of channel-facing residues, which fine tune this dimension and introduce different chemistries.

αHBs are coiled-coil assemblies of polypeptides encoded by heptad sequence repeats, *abcdefg*, with predominantly hydrophobic residues at *a, d, g* and *e* (Supplementary Fig. 1)[27,29]. Four such repeats give stable assemblies with channels ≈4 nm in length. The *a* and *d* sites define the channel and contribute to the helix-helix interfaces. Open αHBs require combinations of predominantly *a* = Leu/Ile/Met/Val and *d* = Ile/Val[27,29]. The *g* and *e* sites also contribute to the helical interfaces, but substitutions at *g* have the greater impact on oligomer state[30]. Therefore, we kept *e* = Ala in most designs and made *g* = Ala, Asn, Gln, Glu, Ile or Ser to sample oligomer states of 5–8 and internal diameters of ≈5–10 Å[27,34]. Side chains at *b* and *c* were made complementary pairs of Glu and Lys or Arg to introduce favorable and solubilizing inter-helical charge-charge interactions. The *f* positions are largely redundant in defining coiled-coil structure, and were made combinations of

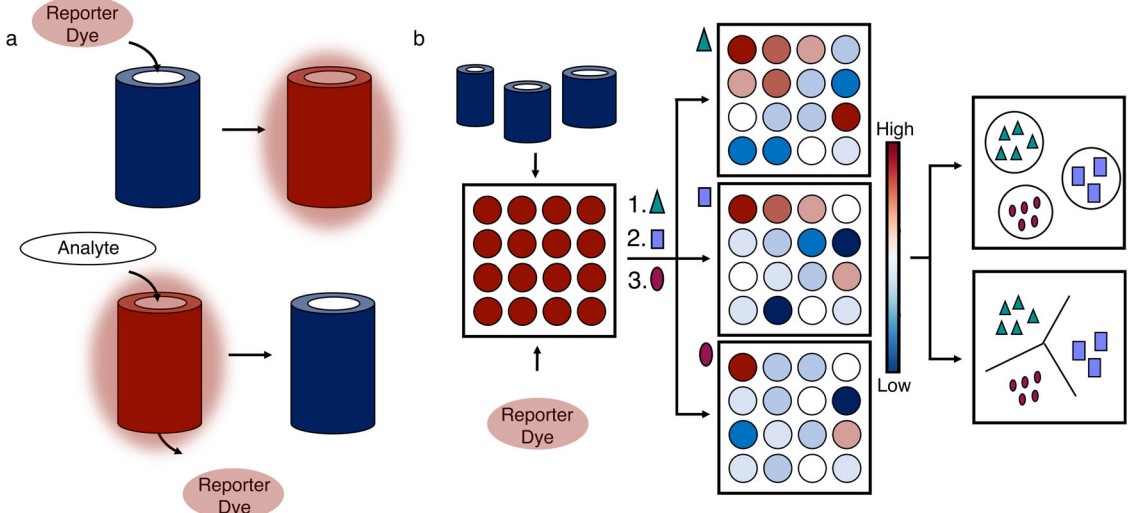

**Fig. 1 | Concept for the de novo designed α-helical-barrel differential sensor.** **a** Top: α-Helical barrels (αHBs) are loaded with an environment-sensitive dye giving a fluorescent signal. Bottom: The dye is displaced by an analyte causing a loss of fluorescence that can be measured. **b** Left: Different αHBs are combined with the environment sensitive dye in multi-well plates. Middle: The resulting array is challenged with different analytes, which can be pure compounds or complex mixtures.

Depending on the relative binding strengths of the dye and the analytes for each αHB, dye is displaced differentially across the array to give a 'fingerprint' for each analyte. Right: Statistical and machine-learning methods are used to classify the different fingerprints and relate them to the analytes. The resulting models can be used as predictive classifiers for naïve samples. See Supplementary Note and Supplementary Figs. 10–12 for more detail on the data analysis and ML pipelines.

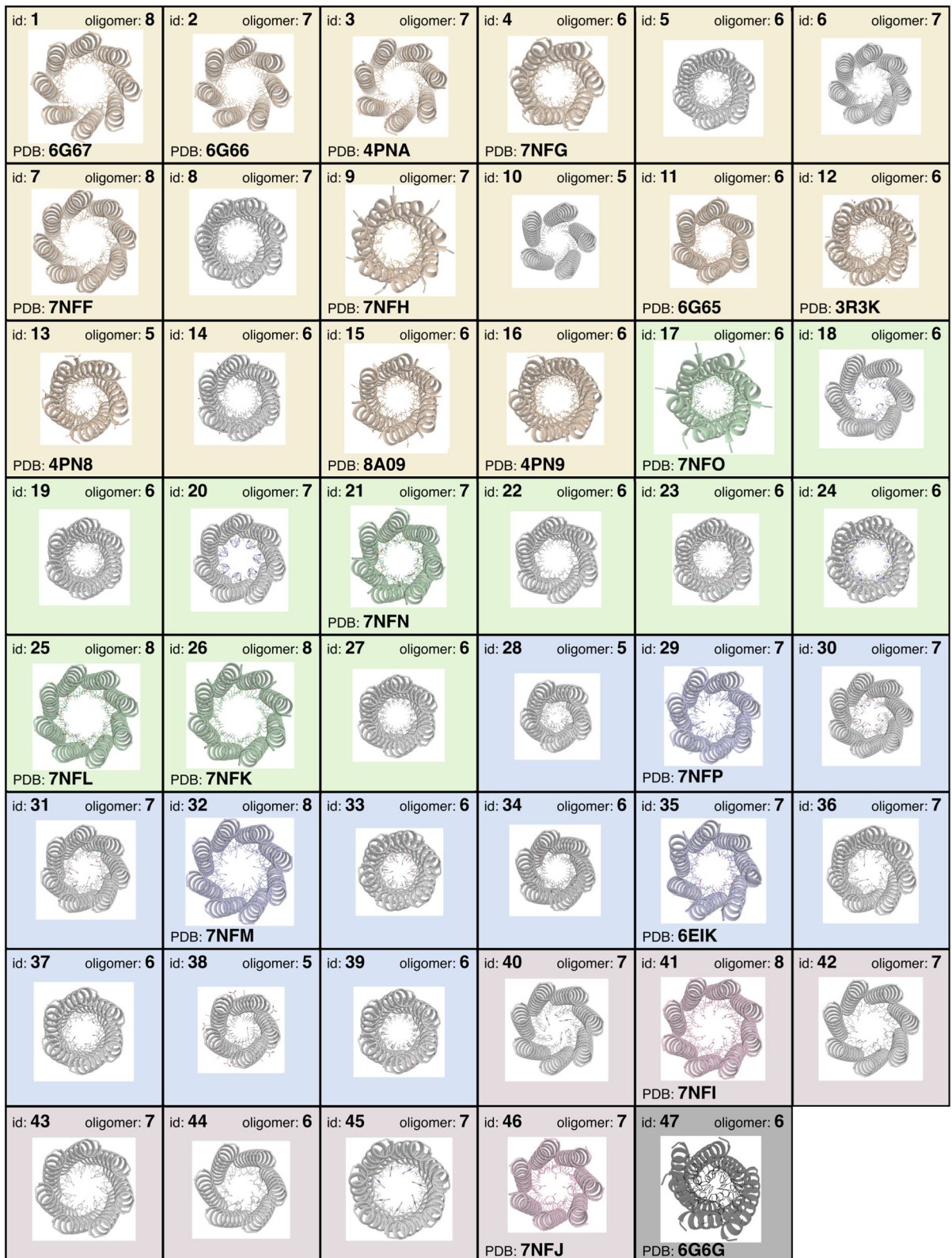

helix-favoring, water-soluble Lys and Gln, with a single Trp introduced as a chromophore to allow for accurate concentration measurements.

Next, we focused on the channel chemistry to allow the binding of a wide variety of small molecules. Despite the requirements for aliphatic residues at *a* and *d*, up to 40% of these can be changed to other side chains without compromising barrel integrity[32]. We introduced mutations at one or two of these sites to generate four groups of αHB

(Fig. 2 and Supplementary Data File 1): Group I had entirely hydrophobic interiors, but with different sizes and shapes of channel; Groups II and III had polar uncharged or polar charged residues, respectively, at specific points along the channel; and Group IV had aromatic residues installed in their channels.

Of these 46 αHBs, 13 have been characterized previously[27–29,34]. The remaining 33 were synthesized by solid-phase peptide synthesis,

**Fig. 2 | Computationally and rationally designed αHBs as arrayed in the sensor.** The four groups of αHB used in the αHB sensor arrays are shaded by group: Group I, hydrophobic (yellow); Group II, polar-uncharged (green); Group III, polar-charged (blue); and Group IV, aromatic (red). Colored models and PDB entry codes are given for those αHB where X-ray crystal structures were obtained; otherwise, the models shown (gray) were built and optimized using CCBuilder2.0[35]. In detail (see Supplementary Table 1), the sixteen Group I peptides included: previous designs for a pentamer, 3 hexamers, 2 heptamers and an octamer all verified by X-ray crystallography;[27,29] single and/or double mutations to Ala and Gly at central *a* and *d* sites to generate larger channels; a single-Pro mutant at the final *d* site to kink and open the C-terminal end of the channel; and a variant with all *a* sites made Met to vary the hydrophobic chemistry used. The eleven Group II peptides comprised: single mutants to Cys, His, Asn, Ser or Thr at *d* sites; and double mutations to His, Asn, Ser or Thr at consecutive *a* and *d* sites. The twelve Group III peptides had positively and negatively charged side chains, Lys and Glu, incorporated either singly or paired at *a* and *d* sites; and a single peptide with Asp at *d*. Finally, seven Group IV designs incorporated single Tyr at *a* or *d* sites, or Trp residues at *a* sites.

purified by HPLC, and confirmed by MALDI-TOF mass spectrometry (Supplementary Figs. 2 and 3 and Supplementary Data File 1). All 33 peptides were highly helical and thermally stable (Supplementary Figs. 4 and 5). By sedimentation-velocity experiments using analytical ultracentrifugation, all formed single discrete species with molecular weights ranging from pentamer to heptamer (Supplementary Fig. 6, Supplementary Table 1 and Supplementary Data File 1). Finally, one third of the designs were crystallized and yielded 12 X-ray crystal structures, all of which were open αHBs with fully accessible channels (Fig. 2; Supplementary Fig. 7, Supplementary Table 2, and Supplementary Data File 2). Where experimental structures were not obtained, the sequences were modelled and optimized as αHBs with the experimentally determined oligomer state (Fig. 2) using computational design[35].

For the final sensing array, we added two controls—a no-peptide blank, and a collapsed hexameric bundle that does not bind DPH[29]—to give a 48-component array (Fig. 2).

## αHB arrays classify small-molecule metabolites and biomarkers

Initially, we tested the αHB sensor array (αSA) using three categories of biological small molecules: amino acids (AAs), carbohydrates (CHOs), and fatty acids (FAs). In each case, five molecules were chosen to maximize chemical variation and biological relevance (Supplementary Fig. 8)[36–38]. For the AAs, we chose Ser, as a small and polar side chain; Val (small hydrophobic); Arg (large, charged, and basic); Glu (large, charged, and acidic); and Trp (large aromatic). For the CHOs, four monosaccharides involved in metabolism—glucose, fructose, mannose and glucosamine—plus the disaccharide maltose were selected. The FAs spanned a range of carbon-chain lengths with and without double bonds: butyric acid (4:0, 4 carbons:0 double bonds); decanoic (capric) acid (10:0); palmitic acid (16:0); oleic acid (18:1); and nervonic acid (24:1).

The full αSA was challenged separately with each of the 15 molecules with 10 repeats for each. Pre-processing of the data (Supplementary Note and Supplementary Figs. 9–11) removed outliers from liquid-handling errors to give ≥45 data points for each type of molecule (Fig. 3 and Supplementary Figs. 12–14). Given that the channels of αHBs are predominantly hydrophobic, we anticipated that FAs would displace more of the reporter dye and give higher signals than the AAs or CHOs. Indeed, with the FAs, every αHB had signal above the control baseline for at least one FA; and almost all αHBs showed full displacement of the reporter dye when challenged with C16:0, C18:1 and C24:1 FAs (Supplementary Fig. 13). The more-polar AAs and CHOs gave markedly different responses (Fig. 3 and Supplementary Figs. 12, 14). For the five AAs, signal was substantially lower across the αSA compared with FAs. However, the large, hydrophobic Trp gave consistently greater signals as expected (Fig. 3a). The low signal was even starker when the αSA was challenged with CHOs, with most αHBs responding similarly. This highlights a known challenge of binding and sensing CHOs in aqueous media[39].

Analysis of the αSA responses for all three types of small molecule, showed that 44 of the 46 αHBs gave consistent readings. Two αHBs (peptide ID 15 and 30) showed greater variance in signal for all analytes, which we attribute to low "loaded" fluorescence intensities, resulting in a smaller signal-to-noise ratio upon challenge with analytes. Spearman's rank correlation coefficients ($\rho$) were calculated for all αHBs for the three types of small molecule (Supplementary Figs. 15–17). As might be expected from the weaker signals for the AAs and CHOs, we observed less correlation between individual αHBs in these challenges due to the low signal-to-noise ratio masking weak dye displacement. By contrast, there were much higher correlations between αHBs in the FA challenges ($\rho > 0.6$ between all αHBs). Nonetheless, these analyses indicated that the αSA could be reduced in size for each application: the lower correlation for polar analytes implies that αHBs not providing signal above noise could be removed; and, conversely, the high correlation with FAs implies that multiple αHBs are providing similar information.

To assess the classification potential of the αSA, ML models were used to differentiate each molecule within its own class (Supplementary Note and Supplementary Figs. 10 and 11). Briefly, six algorithms were tested—Gaussian naïve Bayes, k-nearest neighbors[40], linear discriminant analysis (LDA), an AdaBoost[41] classifier, and two support-vector classifiers with a linear kernel (linear SVC) or with a radial basis function kernel (SVC)[42]—with the aim of selecting the simplest model with the best performance. Training used nested stratified cross-validation and the average accuracy across all folds was calculated. Two dummy classifiers were also applied that assign random class labels to every sample, mimicking random guessing. To optimize the αSA for each challenge, feature analysis was used to identify αHBs that contributed above others to each algorithm. Two methods were used for this: KBest analysis and permutation analysis. The ML algorithms were then run using the identified features to give the final performance metric of the models. Finally, αSA performance (of the full array) was compared to the dummy classifier using a 5 × 2 CV F-test (Supplementary Fig. 11)[43]. The reduced and full αSAs were also compared (with the 5 × 2 CV F-test) to monitor any change in performance from reducing the size of the array.

From principal component analysis (Fig. 3b, Supplementary Fig. 13 and 14), the variance between the FA classes was significantly greater than for the AAs, which was greater than for the CHOs. This was reflected in the classification results (Table 1 and Supplementary Tables 3–5): the FAs were predicted/classified with 100% accuracy from two features/αHBs (with three different ML models); the AAs with 69 ± 16% accuracy from 10 features (Gaussian Naïve Bayes, average ± standard deviation); and the CHOs with 61 ± 23% accuracy from four features (SVC). Clearly, the system performs less well at discriminating within the sets of small, polar analytes. However, these accuracy levels are still significantly above both the dummy classifiers as determined by 5 × 2 CV F-test (Supplementary Tables 3–5). Interestingly, whilst the AAs and FAs showed no significant difference between the full and reduced-feature αSA (*p*-value = 0.60 and 0.38, respectively), the four-feature αSA significantly outperformed the full αSA in the classification of CHOs (*p*-value = 0.029), suggesting the other 42 αHBs are simply contributing noise to the αSA signal.

## αSAs differentiate complex mixtures with high accuracy

To test the possibilities of using αSAs to distinguish complex mixtures[5], we sought to identify different types of tea as a well-characterized mixture used previously in differential sensing[44–46]. For this, we used a smaller αSA of 14 barrels from Classes I – III

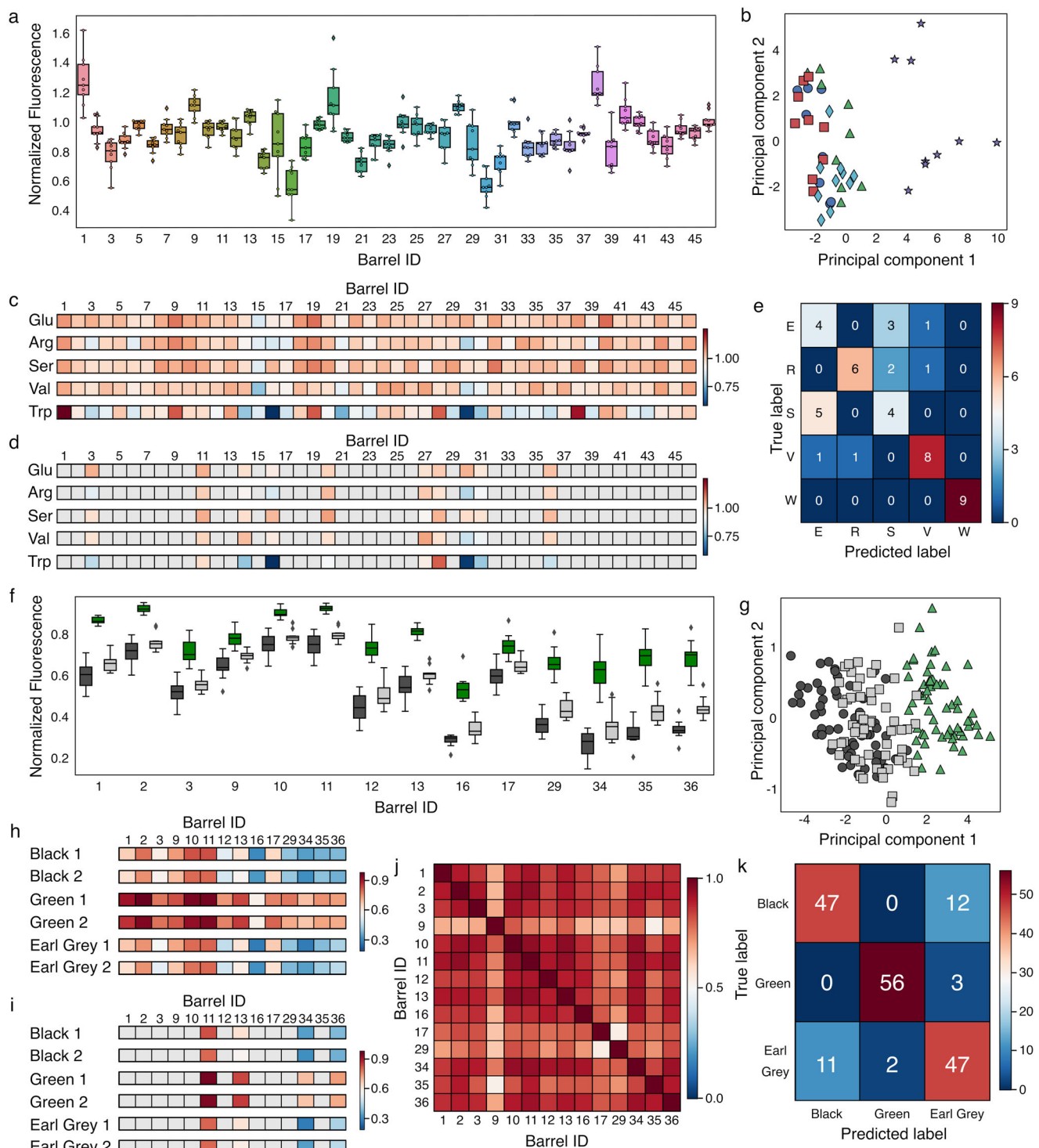

(Figs. 2 and 3, Supplementary Data File 1), which consisted of peptides that had been characterized previously[27–29]. We tested three classes of tea— black tea, Earl Grey, and green tea—and chose 10 brands for each (Supplementary Table 6). We collected 6 replicates for each brand, resulting in 178 tea fingerprints after outlier removal to train the ML algorithms.

Visual inspection of the fluorescence data and principal components of the fingerprints revealed structure in the data, with green tea forming a distinct group and black tea and Earl Grey tea overlapping (Fig. 3f, g). Earl Grey is a black tea with an essential oil from the rind of the bergamot orange added. So, it is reasonable that the fingerprints are similar (Fig. 3h, i,

Supplementary Fig. 18). Correlation coefficients (Fig. 3j) were relatively high between all 14 αHBs, and feature analysis reduced the αSA further to four peptides (Fig. 3i).

The six classifiers introduced above were trained to identify samples as black, Earl Grey, or green tea using nested stratified cross-validation, and the average accuracy across all folds calculated (Table 1 and Supplementary Table 7). All models except AdaBoost showed similar performance ranging ≈75 – 85% predictive accuracy, significantly above the dummy classifiers (*p*-value = 2 × 10⁻⁶, Supplementary Table 7). The confusion matrix from these tests confirmed the trend observed in the principal component analysis plot: the classifiers were highly accurate for

**Fig. 3 | Differentiating amino-acid biomarkers and a complex mixture using the αSA. a** Min–max scaled fluorescent signals from the αSA with tryptophan. Values are normalized relative to: 1, for the αHB and the reporter dye with no analyte; and 0, for the dye alone. Data shown corresponds to n = 9 independent samples. Boxes show the interquartile range with the median presented as a line. Whiskers show 1.5× interquartile range, or the range if a smaller value. Outliers are shown as diamonds. **b** Principal component analysis of the 5 amino acids: glutamate, blue circles; arginine, green triangles; serine, red squares; valine, cyan diamonds; and tryptophan, purple stars. **c** Representative dye-displacement data for each analyte in the AA group. αHB ID is shown above each fingerprint. In these cases, min–max scaled dye displacement is colored from dark red (less displacement) to dark blue (more displacement) according to the respective heat maps (right-hand side of each panel). Each fingerprint corresponds to the median signal across all repeats for each AA. **d** The 10 features selected to take forward to classification. Color scheme as in **c**, αHBs not selected are colored gray. **e,** Confusion matrix generated from the classification of AA samples using the Gaussian Naïve Bayes algorithm with nested stratified cross-validation. Here coloring scheme is from dark red (all prediction) to dark blue (no predictions) according to the heat map (right-hand side).

**f** Min–max scaled fluorescent signals from the αSAs challenged with different teas. Values are normalized as in (**a**). Black tea, black bars; green tea, green bars; Earl Grey tea, gray bars. Data shows corresponds to 178 independent samples (n = 59 black, n = 59 green and n = 60 Earl Grey). Box and whiskers are presented as in **a**. **g** Principal component analysis of the 178 brewed tea samples: black teas, black circles; green teas, green triangles; Earl Grey teas, gray squares. **h** Representative dye-displacement data for select tea samples (full range shown in Supplementary Figure 18). αHB ID is shown above each fingerprint. Color scheme as in (**c**). In this case, each fingerprint corresponds to the median signal of the 6 independent tea samples for each brand of tea. **i** The 4 features selected to take into classification. Color scheme as in (**c**), αHBs not selected are colored gray. For visualization purposes, the fingerprints in **h** and **i** are the median fingerprints from the 6 independent repeats for each tea brand rather than the 178 individual fingerprints. **j** Spearman coefficients of the αHBs in the αSA for the tea fingerprints. Color scheme is from strong correlation (dark red) to no correlation (dark blue) according to the heat map (right-hand side). **k** Confusion matrix generated from predictions of tea samples using the SVC algorithm with nested stratified cross-validation. Color scheme as in (**e**). Source data are provided as a Source Data file.

**Table 1 | Performance summary of the αSA for different analytes and complex mixtures**

| Analyte/Mixture | Algorithm[a] | Data set size[b] | Number of Features | Accuracy[c] (%) | Precision[c] (%) | F1 Score[c] (%) |
|---|---|---|---|---|---|---|
| Amino acids | Gaussian Naive Bayes | 45 | 10 | 69 ± 16 | 73 ± 20 | 69 ± 17 |
| Carbohydrates | SVC | 48 | 4 | 61 ± 23 | 61 ± 29 | 58 ± 25 |
| Fatty acids[d] | Gaussian Naive Bayes | 45 | 2 | 100 ± 0 | 100 ± 0 | 100 ± 0 |
| Tea | SVC | 178 | 4 | 84 ± 10 | 87 ± 9 | 84 ± 10 |
| NASH (2-way) | SVC (linear) | 41 | 5 | 90 ± 5 | 93 ± 4 | 90 ± 6 |
| NASH (3-way) | LDA | 42 | 4 | 74 ± 15 | 80 ± 11 | 74 ± 14 |

[a]LDA – linear discriminant analysis. SVC – support vector classification.
[b]After the required pre-processing as detailed in the Supplementary Methods.
[c]Mean value from all k-folds ± standard deviation.
[d]K-Nearest neighbors and SVC also gave 100% accuracy.

green tea (97%) but performed less well with the more-similar black and Earl Grey tea fingerprints (87% and 84% respectively, Fig. 3k). 10/10 of the green tea brands were correctly predicted, compared with 9/10 Earl Grey brands and 8/10 black tea brands (Supplementary Table 8).

### Sera can be analyzed and classified using the αSA

Next, we turned to medical samples that might be distinguished due to the αHBs binding lipids. Fatty acids and lipids are a significant proportion of the small molecules in blood, and the plasma lipidome is affected by many disease states[38,47]. For instance, in non-alcoholic fatty liver disease (NAFLD) fatty acid and lipid metabolism is altered in patients[48,49]. NAFLD has multiple stages—steatosis, non-alcoholic steatohepatitis (NASH), fibrosis, and cirrhosis—and is reversible if diagnosed early[50]. Current diagnosis requires an ultrasound or biopsy, creating a need for simple in vitro diagnostics[50]. We asked if the αSA would be suitable for this.

Serum samples from 14 patients diagnosed with NASH were compared with sera from 28 donors without NASH. All patients had co-morbidities (Supplementary Table 9), including coronary artery disease (CAD) in all 14 NASH patients. Therefore, 14 CAD patients were also analyzed to discriminate between indirect changes in the serum lipidome. The subjects were all female, and they were matched in age and BMI as closely as possible. Each of the 42 sera samples (14 NASH samples, 14 CAD samples, and 14 control samples) were measured four times, each with four technical replicates, using the αSA with 46 αHBs. A median value was calculated for the 16 repeats of each sample. The data were preprocessed and analyzed as above (Supplementary Note).

Principal component analysis of the NASH and non-NASH data showed separation, albeit with some overlap (Fig. 4b). Again, correlation coefficients of the αSA were relatively high for similar αHBs

(Fig. 4e, f, Supplementary Fig. 21); namely, the larger hydrophobic αHBs (IDs 1–10), the double polar residue mutants (IDs 20–23), the charged αHBs (IDs 31–36), and the aromatic αHBs (IDs 40–45). Next, applying our αSA ML pipeline, all 6 algorithms performed well with LDA and linear SVC giving the highest performance with 90 ± 6% average F1 scores in both cases using five features (Table 1, Supplementary Table 10). When all three classes were considered—NASH, CAD, and the control group—the model performance decreased to 74% ± 14% (LDA, 4 features; Supplementary Fig. 22, Supplementary Table 11), but was still significantly better than the dummy classifiers (p-value = 0.004). However, when incorrect, the model predicted NASH and CAD samples as controls rather than the other disease category (Supplementary Fig. 22). This implies that CAD and NASH are responsible for the predominant signal from the αSA, with the signal from the control group overlapping these. To probe this further, PCA was performed on the non-obese sera samples (i.e., BMI < 30; Supplementary Figure 23). The resulting 2D plot indicates that the groups remain separable, demonstrating the αSA is picking up NASH- and CAD-specific signals. This demonstrates that the αSA is able to differentiate samples from donors with different disease presentations, rather than a disease state in general. Thus, in a 2-class problem with CAD combined with the other non-NASH samples, the more-subtle specific NASH signals can be learnt by the ML algorithms.

We note that FAs will be associated with albumin in blood, and that this may well affect the available free FAs for detection by the αSA.

### De novo αHBs for designer sensors

Differential sensors depend on the combined response of many low-specificity receptors when challenged with different molecules. Our study indicates that αSAs act similarly, and that αHBs are analogous to olfactory GPCRs and other synthetic receptor-based systems in this

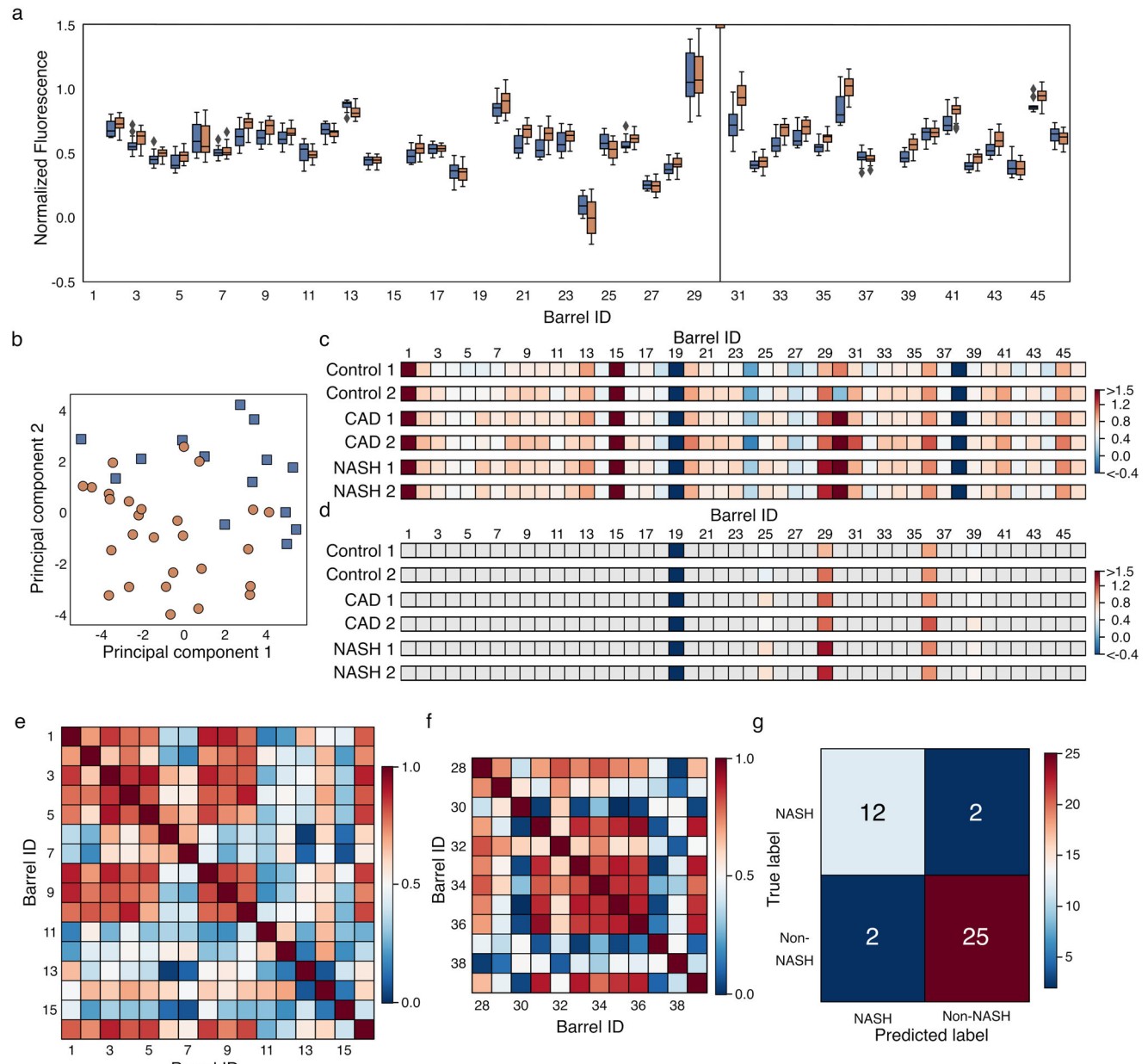

**Fig. 4 | Challenging the αSA with diagnostically relevant samples. a** Min−max scaled fluorescent signals from the αSAs challenged with different NASH sera samples: NASH, blue; Non-NASH, orange. Values are normalized relative to: 1, for αHB and the reporter dye with no analyte; and 0, for dye alone. Values shown are between 1.5 and −0.5 for clear visualization, full data range is shown in Supplementary Fig. 19. Data corresponds to 41 independent samples ($n = 14$ NASH, $n = 27$ Non-NASH) that were each measured 4 times (technical repeats) to give a median measurement for each sample. Boxes show the interquartile range with the median presented as a line. Whiskers show 1.5× interquartile range, or the range if a smaller value. Outliers are shown as diamonds. **b** Principal component analysis of the 41 sera samples: NASH, blue squares; Non-NASH, orange circles. **c** Median dye-displacement data for select NASH sera samples. αHB ID is shown above each fingerprint. In these cases, min-max scaled dye displacement is colored from dark red (less displacement) to dark blue (more displacement) according to the respective heat maps (right-hand side of each panel). Data values are limited to between 1.5 and −0.4 for clear visualization purposes. Each fingerprint is the median value from 16 repeats of each serum sample (4 independent repeats, each consisting of 4 technical replicates). **d** The 5 features selected for classification. Color scheme as in (**c**), αHBs not selected are colored gray. Spearman's rank coefficients of class I (**e**) and class III (**f**) αHBs in the αSA for the NASH fingerprints. Color scheme is from strong correlation (dark red) to no correlation (dark blue) according to the heat map (right-hand side). **g** Confusion matrix generated from predictions of NASH sera samples using the linear SVC algorithm with nested stratified cross-validation. Here the coloring scheme is from dark red (all prediction) to dark blue (no predictions) according to the heat map (right-hand side). Source data are provided as a Source Data file.

respect. Importantly, by applying feature importance analysis methods in the αSA ML pipeline, the most-discriminative αHBs for a given challenge can be identified. For the datasets we have collected and described here, the signal can be captured by just 2–10 barrels.

To investigate this further, for the four challenges that employed the whole 46-barrel array−i.e., excluding the analysis of the teas−each αHB was ranked for importance by three feature-selection methods:

KBest analysis, an ExtraTrees classifier, and permutation analysis (Fig. 5a, Supplementary Fig. 24). With some differences (Supplementary Note), for each challenge, the top-five most-important αHBs were generally consistent between the three methods. Interestingly, however, in each challenge a different αHB was identified as the most important by at least 2/3 of the feature-selection methods: barrel ID 16 for AAs; ID 17, FAs; ID 41, CHOs; and ID 39, NASH/non-NASH. To

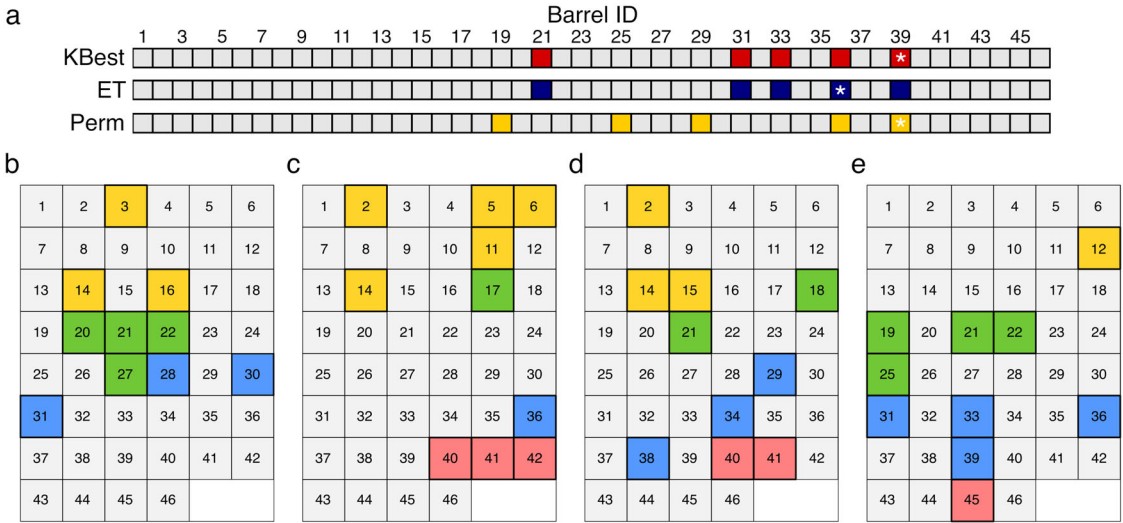

**Fig. 5 | Features/αHBs that contribute most to the αSA signal for different classifications. a** Feature importance of the 46 αHBs in the αSA in the differentiation of NASH and non-NASH sera samples. The top 5 features as determined by KBest analysis, ExtraTrees (ET) and permutation (perm) analysis are highlighted (red, blue and gold, respectively). The highest ranked αHB from each feature selection method is highlighted (*). The 10 αHBs that contribute most to signal in challenges with AAs (**b**), FAs (**c**), CHOs (**d**), and 2-way NASH sera classification (**e**). αHB rankings are taken from the combined rank of all three feature selection methods. Color scheme for **b**–**e**: Hydrophobic αHBs, yellow; polar mutations, green; charged mutations, blue; aromatic mutations, red.

explore this more deeply, the most-important features of the whole αSA were compared across the four challenge classifications. To do this, ranks from the three feature-selection methods were summed to give an overall αHB ranking across all feature-selection methods for each problem.

The top 10 αHBs were compared. Intriguingly, this revealed that each classification problem required a different subset of αHBs in the αSA (Fig. 5b–e). However, the features that contributed the highest signal to each αSA response are not necessarily the αHBs that interact most strongly with the analyte/mixture (Figs. 3a and 4a; Supplementary Figs. 12–14). For example, feature selection with the NASH/non-NASH data revealed αHBs with relatively little dye displacement, and, thus, smaller signal losses (typically between 0.5 and 1.0, Fig. 4a); whereas, other αHBs showed greater dye displacement (e.g. ID 18, 24, 27 and 44). Thus, it is the difference between the sample classes that is more important than the overall binding affinity of the challenge in dictating the αSA performance. This is consistent with requirements for differential sensing where numerous low-affinity interactions contribute to the sensor.

Focusing on the small molecules (Fig. 5b–d), the αSA signal from the more-polar AAs and CHOs is dominated by the polar and charged Group II and III αHBs (Fig. 2). Conversely, the FAs generate signal through interactions with the more-hydrophobic and aromatic-containing channels (Group I and IV, Fig. 2). This correlates with our design rationale and understanding of these de novo designed peptide assemblies[28,32]. Moreover, there was little overlap between the optimal αSA required for the AA, CHO, FA, and NASH/non-NASH classifications. This indicates that the αHBs underpinning αSA can be designed towards a specific application. Thus, we envisage that a master array of rationally designed αHBs in combination with an ML pipeline could be used to identify subsets of αHBs as bespoke mini-arrays for different applications.

In summary, we have presented a robust and adaptable differential sensor, the αSA, using de novo designed peptide assemblies as its receptor components. The designed peptides form α−helical barrels (αHBs) that are mutable and bind a range of small molecules in their channels. In these respects, they are analogous to the GPCRs of mammalian olfactory systems and to other synthetic receptor-based differential sensors. Moreover, given their synthetic accessibility,

water solubility, hyperthermostability, and our ability to tune channel size and chemistry, we contend that αHBs are ideal components for differential sensors. The αSA that we have made from these de novo peptides differentiates amino-acid, carbohydrate, and fatty acid biomolecules above baseline and without prior optimization. Furthermore, complex mixtures and clinically relevant samples can be classified and predicted, highlighting the potential function of the αSA platform in diagnostics. The αSA utilizes a machine-learning pipeline that allows users to spot check a wide range of algorithms to determine the underlying performance. Through this, feature selection can identify subsets of αHBs to make bespoke sensor arrays. We envisage the platform being developed into sensors for biotechnological, environmental, and medical diagnostics applications.

## Methods

### Ethical statement
Serum samples from donors with NASH, CAD and corresponding healthy controls were purchased from the commercial biobank Proteogenex Inc. The protocols for obtaining samples were approved by the Ethics committee of the host organization (PG-ONC 2003/1, 9/1/2020), with all donors signing informed consent documentation.

### General
Peptide sequences and ID number can be found in Supplementary Data File 1. Relevant characterization data for previously published peptides are available[27–29,34]. Fmoc-amino acids were purchased from Biosynth Carbosynth or Cambridge Reagents. All other chemicals were purchased from Merck or VWR. Peptide biophysical characterization was performed in phosphate buffered saline, 8.2 mM sodium phosphate, 1.8 mM potassium phosphate, 137 mM sodium chloride, 2.7 mM potassium chloride at pH 7.4 unless otherwise stated.

### Peptide synthesis, purification and characterization
Peptides were synthesized using standard Fmoc solid-phase peptide synthesis methods, on a microwave assisted Liberty Blue (CEM) peptide synthesizer. Peptides were purified by reverse phase HPLC (Luna C-18(2) column) and confirmed as the target sequence by analytical HPLC and MALDI-TOF spectrometry. CD spectra were measured with 10 μM peptide at 20 °C between 200 and 260 nm in PBS on a Jasco

J-810 or J-815 spectropolarimeter in a 5 mm cuvette, and data collected using Spectra Manager. Thermal denaturation measurements were performed between 5 and 95 °C at 222 nm with 10 µM peptide in PBS in 5 mm cuvettes. AUC SV measurements were performed on a Beckmann XL-A with 150 µM peptide at 20 °C in PBS at 50000 rpm. Data was collected with Proteome Lab XL-A, and analyzed with SEDFIT.

## X-ray crystal structure determination

Lyophilized peptides were dissolved in deionized water to concentrations of ≈10 mg/mL. Vapor diffusion trials were performed at 20 °C using commercial screens: JCSG-Plus™, Morpheus®, PACT Premier™, ProPlex™ and Structure Screen 1 + 2. Prior to freezing, crystals were soaked in cryoprotectant consisting of their respective crystal screen with 25% v/v glycerol. Final crystallization conditions for all peptides are given in Supplementary Table 2. Data was collected at Diamond Light Source on beamlines I02, I04 and I04-1. Data were processed using automated methods: Xia2 pipelines[51], which ports data through DIALS[52] or MOSFLM[53] to POINTLESS and AIMLESS[54] as implemented in the CCP4 suite[55], or XDS to XSCALE[56]. Structures were solved using molecular replacement from poly-alanine models as determined by the relevant Matthew's Coefficient, using PHASER[57]. Final models were obtained after subsequent refinement rounds using PHENIX Refine[58] or Refmac5[59] and model building in COOT[60]. Solvent-exposed atoms lacking map density were modelled at zero occupancy. Data collection and refinement statistics are provided in Supplementary Data File 2.

## α-Helical barrel sensor array assay

For the full-peptide array, peptides (20 µM final concentration) were premixed in 2× HEPES buffered saline (50 mM HEPES, 200 mM NaCl, pH 7) and 1,6-diphenyl-1,3,5-hexatriene (DPH; 2 µM, 10% v/v DMSO final concentration) and dispensed to 384-well microplates using a Tecan Freedom EVO® liquid handling station. From the 47 different peptide solutions and the dye control, 10 µL was added to each well, respectively, to create eight 48-array patterns across the plate. Once plates were produced, they were stored at −80 °C until usage. Analytes were dispensed in 10 µL aliquots using a TECAN Freedom EVO® liquid handling station or a Multidrop Combi liquid handler giving a 1:1 peptide/dye to analyte ratio in each well with a final concentration of 10 µM peptide, 1 µM DPH, 5% v/v DMSO and 1× HBS (25 mM HEPES, 100 mM NaCl, pH 7). Microwell plates were analyzed using a CLARIOstar plate reader ($\lambda_{ex} = 350 \pm 15$ nm, $\lambda_{em} = 450 \pm 20$ nm).

For the analysis of the small molecules, all analytes were dissolved in deionized water (20 mL) at the desired concentration: amino acids (AA) and carbohydrates (CHO) at 20 mM, fatty acids (FA) at 20 µM. Fatty acids required 5% v/v DMSO in the stock solutions for solubility. Independent samples were prepared for each repeat ($n = 10$). Small molecule samples were dispensed onto prepreprared 384-microwell plates (10 mL) using a Tecan Freedom EVO® liquid handling station. This gave a final concentration of 10 mM for the AAs and CHOs in each well, and 10 µM for the FAs (with a final concentration of 7.5% v/v DMSO for the FAs).

For the analysis of the tea samples, a total of thirty brands of teabags (comprising 10 black, 10 Earl Grey, and 10 green tea varieties, see Supplementary Table 6) were purchased. For the preparation of brewed tea samples, where applicable, strings and labels were removed from tea bags. A single tea bag was placed in boiled deionized water (250 mL), and the tea allowed to brew for 5 min with stirring. After this time, 1 mL of the tea solution was removed, and diluted 1:10 with deionized water before snap freezing in liquid nitrogen and stored at −80 °C. Fresh tea samples (from the same batch/box of teabags) were prepared for each experimental replicate ($n = 6$). Tea samples were dispensed onto prepreprared 384-microwell plates (15 µL) using a Multidrop Combi liquid handler.

For the smaller array of 15 peptides (used to analyze the tea samples), 384-well microplates were prepared using a Tecan Freedom EVO® liquid handling station. Deionized water (6 µL), 10× HBS (250 mM HEPES, 1 M NaCl, pH 7, 3 µL), DPH (10 mM, 50% v/v DMSO, 3 µL) and peptide (100 µM, 3 µL) were added to each microwell giving 24 16-array patterns of 15 µL aliquots (2× HBS, 20 µM peptide, 2 µM DPH, 10% DMSO) across the plate. Once plates were produced, they were stored at −80 °C until usage. Samples were dispensed in 15 µL aliquots using a Multidrop Combi liquid handler giving a 1:1 peptide/dye to analyte ratio in each well with a final concentration of 10 µM peptide, 1 µM DPH, 5% v/v DMSO and 1× HBS (25 mM HEPES, 100 mM NaCl, pH 7). Microwell plates were analyzed using a CLARIOstar plate reader ($\lambda_{ex} = 350 \pm 15$ nm, $\lambda_{em} = 450 \pm 20$ nm).

For the analysis of NASH, CAD and control sera, 42 1 mL samples purchased from Proteogenex (Supplementary Table 9) were thawed at rt for 30 minutes, aliquoted into 50–100 µL fractions, and re-frozen at −80 °C, where they were stored until required. On the day of analysis, one aliquot of the required serum sample was thawed at rt for 30 min. 40 µL serum sample was added to 8 mL deionized water, resulting in a final serum concentration of 0.5% v/v. Following dilution, the sera were analyzed immediately by dispensing into prepared 384-well microplates (10 µL 0.5% v/v serum sample was dispensed into each well containing 10 µL αHB-DPH mix, resulting in a final serum concentration in each well of 0.25% v/v) using a Multidrop Combi liquid handler. Each sample was analyzed on four separate microwell plates ($n = 4$), each time using a fresh aliquot of the same sample. Therefore, upon analysis, each serum sample had undergone two freeze-thaw cycles.

## Data processing and machine learning analysis

Feature selection and machine learning algorithms were implemented using the open-source Python package, scikit-learn[61]. Two-sided 5×2 CV F-tests were implemented with MLxtend[62].

The raw fluorescent data from the α-sensor array (αSA) assay is min–max scaled using Eq. 1:

$$\text{Normalized data} = \frac{X - \text{Min}_{A+F}}{\text{Max} - \text{Min}_F} \tag{1}$$

where X is the fluorescent output of each α-helical barrel (αHB) with analyte and DPH; $\text{Min}_{A+F}$ is the signal of the analyte with DPH (to correct for autofluorescence); Max is the value of αHB and DPH; and $\text{Min}_F$ the value of DPH alone. Data were converted to dataframe format and technical repeats across the same plate, and different plates if necessary, were averaged by calculating the median. Data outputs are generated for visual inspection to highlight potential anomalous plates. Outliers were identified by a generalized ESD test[63,64] to give the machine learning (ML) dataset. Data outputs are generated again for visual inspection once outliers have been removed.

Six ML algorithms—Gaussian Naïve Bayes, K-nearest neighbors[40,65], linear discriminant analysis, support vector classification (linear and radial basis function kernel)[42] and an AdaBoost classifier[41]—were trained using nested stratified k-folds cross-validation and compared to two dummy classifiers (which mimic random guessing). Feature importance analysis (KBest analysis, an ExtraTrees classifier and permutation analysis) was performed for all datasets. Models trained using the readings measured for all peptides were compared to models trained using the readings from a reduced number of peptides selected by either KBest or permutation analysis. A two-sided 5 × 2 CV F-test[43,66] was used to compare the performance of the reduced αSAs to the full αSA of 46 peptides, and to compare the performance of the full aSA to the dummy classifiers.

## Statistics and reproducibility

No statistical methods were used to predetermine sample size. The details for number of repeats and excluded data for each specific dataset are listed below.

**Fatty acids.** Ten independent solutions were made for each of the five analytes. Each solution was freshly made, with a final concentration of 10 µM. Four technical replicates of each solution were measured using the sensor array, and were averaged by taking the median. This resulted in a dataset of 50 median fingerprints, which after removal of outliers via a generalised ESD test was reduced to 45. Outlier exclusion threshold (i.e., p value) = 0.05; drop threshold (i.e., minimum number of outlier readings required to exclude a fingerprint) = 2. The class distribution was: 10 butanoic acid; 10 decanoic acid; 8 palmitic acid; 9 oleic acid; 8 nervonic acid.

**Amino acids.** The same as for the FAs, except that the final concentration of each solution was 10 mM. The class distribution was: 8 glutamate; 9 arginine; 9 serine; 10 valine; 9 tryptophan.

**Carbohydrates.** The same as for the FAs, except that the final concentration of each solution was 10 mM, and the dataset size after the removal of outliers via a generalized ESD test was 48 fingerprints. The class distribution was: 10 fructose; 10 glucose; 9 glucosamine; 9 maltose; 10 mannose.

**Tea.** Six fresh cups of tea (using different teabags from the same box) were made for each of the 30 brands of tea. Twenty technical replicates were measured using the αSA and were averaged by taking the median. This resulted in a dataset of 180 median fingerprints, which after removal of outliers via a generalized ESD test was reduced to 178. Outlier exclusion threshold (i.e., p value) = 0.05; drop threshold (i.e., minimum number of outlier readings required to exclude a fingerprint) = 2. The class distribution was: 59 black; 59 green; 60 Earl grey.

**NASH sera.** Forty-two serum samples from patients with and without NASH were obtained from a commercial biobank. Four aliquots were taken from each sample, and four technical replicates of each aliquot were measured using our sensor array. Accordingly, 16 fingerprints were measured for each sample. We calculated the median of these 16 replicates to obtain a dataset of 42 fingerprints, which after removal of outliers via a generalized ESD test was reduced to 41 fingerprints (two-way analysis)/no outliers were identified, hence the dataset retained all 42 fingerprints (three-way analysis). Outlier exclusion threshold (i.e., p value) = 0.02; drop threshold (i.e., minimum number of outlier readings required to exclude a fingerprint) = 2. The class distribution for the two-way analysis was: 14 NASH; 27 No-NASH. The class distribution for the three-way analysis was: 14 NASH; 14 CAD; 14 control. Two methods of class balancing—resampling of the smaller class and SMOTE—were tested as part of the nested CV loop for the two-way NASH analysis, and neither was found to lead to a noticeable improvement in model performance. Consequently, the results presented are from a model trained without class balancing. No other covariates were analyzed, and no sex or gender analysis was carried out as the conclusions of this study relate to the performance of the peptide assemblies in the differential sensing technology and their ability to distinguish known samples.

### Reporting summary
Further information on research design is available in the Nature Portfolio Reporting Summary linked to this article.

## Data availability
The αSA data (amino acids, fatty acids, sugars, tea and sera samples), mass spectrometry, circular dichroism and analytical centrifugation data generated in this study are provided as Source Data. The coordinate and structure factor files for peptide ID 4, 7, 9, 15, 17, 21, 25, 26, 29, 32, 41 and 46 have been deposited in the Protein Data Bank with accession codes "7NFF", "7NFG", "7NFH", "7NFI", "7NFJ", "7NFK",

"7NFL", "7NFM", "7NFN", "7NFO", "7NFP" and "8A09". Source data are provided with this paper.

## Code availability
All data and scripts for data processing, model training and model validation, including annotated Jupyter notebooks are available here: https://github.com/woolfson-group/array_sensing (https://doi.org/10.5281/zenodo.7431140)[67].

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

## Acknowledgements

We thank Drs. Murray Brown (GSK) and Andy Boyce (Rosa Biotech) for discussions at the early stages of and throughout the project, respectively. W.M.D., J.M.F., G.G.R., L.L., C.W.W. and D.N.W. were funded by a European Research Council Advanced Grant (340764) and a subsequent European Research Council Proof of Concept Grant (787173). J.M.F., L.L. and D.N.W. were also funded by the BBSRC/EPSRC Synthetic Biology Research Centre, BrisSynBio (BB/L01386X/1). G.G.R. was also supported by the European Union's Horizon 2020 research and innovation programme under the Marie Sklodowska-Curie grant agreement No 88899. L.L. and D.N.W. were also supported by the Elizabeth Blackwell Institute, University of Bristol, with funding from the University's alumni and friends, and a BrisSynBio Flexible Talent Mobility Award (BB/R506539/1). A.J.B., J.A.C., F.J.O.M. were supported by the Bristol

Chemical Synthesis Centre for Doctoral Training funded through the EPSRC (EP/G036764). D.A.S. and K.L.S. were supported by the South West Biosciences Doctoral Training Partnership through the Biotechnology and Biological Sciences Research Council (BB/M009122/1). K.L.S., F.J.O.M. and D.N.W. were also supported by the BBSRC (BB/R00661X/1). We thank the University of Bristol School of Chemistry Mass Spectrometry Facility for access to the EPSRC-funded Bruker Ultraflex MALDI-TOF instrument (EP/K03927X/1) and BrisSynBio for access to the BBSRC-funded BMG Labtech Clariostar Plate Reader and Tecan Freedom EVO 150 liquid handling platform (BB/L01386X/1). We would like to thank Diamond Light Source for access to beamlines I04, I04-1 and I24 (Proposal 12342 & 23269), and for the support from the macromolecular crystallography staff.

## Author contributions

W.M.D. and K.L.S. contributed equally. D.T. and D.N.W. conceived the project. W.M.D., J.M.F., D.A.S., T.L.G., C.W.W., and D.N.W. designed the experiments. W.M.D. designed the peptide array. W.M.D., G.G.R., A.J.B., J.A.C., G.D., and F.J.W. characterized the individual peptides. W.M.D., D.A.S. and L.L. performed the small-molecule assays. W.M.D., J.M.F., D.A.S., and U.O. performed the complex mixture assays. K.L.S. and C.W.W. wrote the ML protocols and performed the analysis. G.G.R., A.J.B., and F.J.O.M. collected X-ray diffraction data, and W.M.D., G.G.R., A.J.B., F.J.O.M., and R.L.B. solved X-ray crystal structures. W.M.D., U.O., and C.W.W. wrote the liquid handling robot protocols. W.M.D., K.L.S., C.W.W. and D.N.W. wrote the paper. All authors have read and contributed to the preparation of the manuscript.

## Competing interests

W.M.D., J.M.F., G.G.R., D.A.S., C.W.W., and D.N.W. are co-inventors on patents WO2019048859 and WO2020178595 covering the use of αHBs as de novo sensors, which are licensed to Rosa Biotech of which J.M.F., D.A.S. and D.N.W. are founders and W.M.D., G.G.R. and C.W.W. own shares. D.N.W. is a director of Rosa Biotech. J.M.F., T.L.G., D.A.S. and U.O. are employees of Rosa Biotech. All other authors declare no competing interests.
