## [Peer Review File · Nature Communications]

Editorial Note: This manuscript has been previously reviewed at another journal that is not operating a transparent peer review scheme. This document only contains reviewer comments and rebuttal letters for versions considered at *Nature Communications*

REVIEWERS' COMMENTS

Reviewer #2 (Remarks to the Author):

The authors have addressed all of my comments.

I only suggest to replace lean-only by non-obese. A BMI < 30 kg/m² is not "lean".

Reviewer #3 (Remarks to the Author):

The revised manuscript and associated SI is significantly improved and clear. I consider that the current version of the manuscript warrants publication. It is a very robust and interesting piece of work that will be of interest to readers in the peptide/protein engineering and sensing fields.

Reviewer #4 (Remarks to the Author):

I read through the manuscript, supplementary materials and responses to the referees and see all my concerns addressed in sufficient detail. Thus, I recommend publication of this manuscript without further modification.

Key:

Reviewer comments in *blue italic text*;

Our responses in regular text; and

Our changes to the manuscript are in **bold text**.

Referees' comments:

Reviewer #2 (Remarks to the Author):

The authors have addressed all of my comments.

I only suggest to replace lean-only by non-obese. A BMI < 30 kg/m² is not "lean".

Reviewer #3 (Remarks to the Author):

The revised manuscript and associated SI is significantly improved and clear. I consider that the current version of the manuscript warrants publication. It is a very robust and interesting piece of work that will be of interest to readers in the peptide/protein engineering and sensing fields.

Reviewer #4 (Remarks to the Author):

I read through the manuscript, supplementary materials and responses to the referees and see all my concerns addressed in sufficient detail. Thus, I recommend publication of this manuscript without further modification.

Author responses:

We are delighted that all Reviewers are satisfied with the revised manuscript.

As suggested by Reviewer 2, we have removed the reference to lean-only in SI document and replaced it with non-obese:

“Principal component analysis of **non-obese** (BMI<30) NASH, CAD and control patients' sera samples. Color scheme: NASH – blue squares, CAD – green triangles, control – orange circles. Source data are provided as a Source Data file.”